# Relaxin-like Gonad-Stimulating Peptides in Asteroidea

**DOI:** 10.3390/biom13050781

**Published:** 2023-04-30

**Authors:** Masatoshi Mita

**Affiliations:** Department of Biochemistry, Showa University School of Medicine, Hatanodai 8-5-1, Shinagawa-ku, Tokyo 142-8555, Japan; bio-mita@med.showa-u.ac.jp; Tel.: +81-3-3784-8116; Fax: +81-3-3784-2346

**Keywords:** relaxin-like gonad-stimulating peptide, 1-methyladenine, cyclic AMP, G protein-coupled receptor, starfish

## Abstract

Starfish relaxin-like gonad-stimulating peptide (RGP) is the first identified peptide hormone with gonadotropin-like activity in invertebrates. RGP is a heterodimeric peptide, comprising A and B chains with disulfide cross-linkages. Although RGP had been named a gonad-stimulating substance (GSS), the purified peptide is a member of relaxin-type peptide family. Thus, GSS was renamed as RGP. The cDNA of RGP encodes not only the A and B chains, but also signal and C-peptides. After the *rgp* gene is translated as a precursor, mature RGP is produced by eliminating the signal and C-peptides. Hitherto, twenty-four RGP orthologs have been identified or predicted from starfish in the orders Valvatida, Forcipulatida, Paxillosida, Spinulosida, and Velatida. The molecular evolution of the RGP family is in good accordance with the phylogenetic taxonomy in Asteroidea. Recently, another relaxin-like peptide with gonadotropin-like activity, RLP2, was found in starfish. RGP is mainly present in the radial nerve cords and circumoral nerve rings, but also in the arm tips, the gonoducts, and the coelomocytes. RGP acts on ovarian follicle cells and testicular interstitial cells to induce the production of 1-methyladenine (1-MeAde), a starfish maturation-inducing hormone. RGP-induced 1-MeAde production is accompanied by an increase in intracellular cyclic AMP levels. This suggests that the receptor for RGP (RGPR) is a G protein-coupled receptor (GPCR). Two types of GPCRs, RGPR1 and RGPR2, have been postulated as candidates. Furthermore, 1-MeAde produced by RGP not only induces oocyte maturation, but also induces gamete shedding, possibly by stimulating the secretion of acetylcholine in the ovaries and testes. Thus, RGP plays an important role in starfish reproduction, but its secretion mechanism is still unknown. It has also been revealed that RGP is found in the peripheral adhesive papillae of the brachiolaria arms. However, gonads are not developed in the larvae before metamorphosis. It may be possible to discover new physiological functions of RGP other than gonadotropin-like activity.

## 1. Introduction

Reproduction systems (reproduction) of animals are broadly divided into asexual reproduction and sexual reproduction. Asexual reproduction is twice as fast as sexual reproduction, so it is considered that asexual reproduction has an advantage and leaves many offspring. However, most eukaryotes living in the earth leave offspring through sexual reproduction. Sexual reproduction requires complicated procedures; gonads have to differentiate into ovaries and testes, and germ-line cells have to undertake meiosis in addition to normal mitosis to produce eggs and spermatozoa.

In mammals, including humans, growth and development of gonads are stimulated by gonadotropin (GTH) secreted from the pituitary gland. There are two types of GTH in terms of roles, luteinizing hormone (LH) and follicle-stimulating hormone (FSH). Both are glycoproteins containing approximately 20% sugar and consist of α and β subunits with non-covalent binding. FSH plays a role in inducing spermatogenesis in the testis and follicle development in the ovary, while LH induces androgen secretion from the Leydig cells in the testis and ovulation of maturing eggs in the ovary. These two types of GTH are structurally and functionally conserved across vertebrates [1], except for Cyclostomata, such as the hagfish, which has only one type of GTH [2,3].

On the other hand, similar gonadotropin-like glycoproteins have not been found in invertebrates. Despite this, a substance with so-called GTH-like action, which stimulates oocyte maturation and ovulation, has been reported in several species among the invertebrates. For example, the insect hormone parsin in the migratory locust is assumed to be the physiological counterpart of LH and FSH in mammals [4]. Some gonadotropic hormones, such as the egg development neurosecretory hormone (EDNH) [5] from the yellow fever mosquito, the egg-laying hormone (ELH) from sea hares [6], and the androgenic gland hormone (AGH) from the wood louse [7], are known. Recently, insulin-like peptides (Insp) have been reported to be responsible for the regulation of egg maturation in malaria mosquitoes, *Aedes aegypti* [8], and the fruit fly *Drosophila melanogaster* [9].

This review introduces starfish gonadotropin-like active peptide relaxin-like gonad-stimulating peptide (RGP). RGP was previously called gonad-stimulating substance (GSS). Starfish GSS was first recognized in invertebrates as a gonadotropin-like active substance. GSS mediates the final maturation of oocytes or meiotic resumption by acting on the ovarian follicle cells to produce the maturation-inducing hormone (MIH) 1-methyladenine (1-MeAde) [10]. In this sense, GSS is functionally identical to vertebrate LH, especially piscine and amphibian LHs (Figure 1A). Considering the functional similarity between GSS and vertebrate LH, it is very important from an evolutionary point of view to know the regulatory mechanism of reproduction in invertebrates and vertebrates.

Generally, oocyte maturation means the process in which immature oocytes stimulated by a hormone overcome meiotic arrest and become fertilizable [11]. In starfish, GSS first stimulates ovarian follicle cells surrounding the oocyte to produce 1-MeAde. Subsequently, 1-MeAde stimulates oocytes to activate the maturation-promoting factor (MPF), a complex of cdc2 and cyclin B, through receptors on the oocyte surface [12]. Finally, oocytes resume meiosis and complete maturation. Thus, GSS is the “primary trigger” for oocyte maturation and corresponds to an LH-like hormone.

Starfish GSS has a long history. GTH-like activity was first reported in 1959 by American biologists Chaet and McConnaughy in a nerve extract of *Asterias forbesi* [13]. Injecting hot-water extracts of the radial nerve cords induced the release of eggs and spermatozoa from ripe females and males, respectively. The active substance was detected in the coelomic fluid of starfish that were naturally gamete shedding, but no activity was found in the coelomic fluid of animals not discharging gametes [14,15]. Thus, it was considered that the active substance contained in the nerve extract was a hormone. The active substance was first named a gamete-shedding substance (GSS) [13], then renamed gonad-stimulating substance (GSS) because the action is indirect [16,17,18]. It took 50 years from the initial finding of Chaet and McConnaughy (1959) before GSS was finally purified from blue-bat star *Patiria* (synonym, *Asterina*) *pectinifera* and its chemical structure identified [19].

## 2. Identification of Starfish GTH-like Active Hormone

Purified GSS from the radial nerve cords of *P. pectinifera* is a heterodimeric peptide with a molecular weight of 4740 Da, comprising an A chain (24 amino acids) and a B chain (19 amino acids) with disulfide cross-linkages, two interchains between the A and B chains and an intrachain within the A chain [19] (Figure 2A). The size of the GSS gene is 3896 base pairs (bp), and it consists of two exons of 208 bp and 2277 bp flanked by one intron (1411 bp) [20]. The size of the mRNA transcribed from this gene is 2486 nucleotides (nt), of which the open reading frame is 351 nt. This indicates that 14% of the total mRNA is translated into peptides. The translated peptide consists of 116 amino acids; the signal peptide, B chain, and C-peptide are arranged from the N-terminus, and the A chain is located in the C-terminal end. The C-peptide of 44 amino acids is sandwiched between the A and B chain. There are three proteolytic cleavage sites (Lys-Arg) in the C-peptide, of which two are located at either end of the peptide, and one is in the middle (Figure 3A). Thus, GSS is first translated from mRNA as a precursor containing the signal and C-peptides (prepro-GSS). After formation of disulfide bonds between the A and B chains, and within the A chain, an active GSS molecule is generated by eliminating the signal and C-peptides.

Furthermore, when chemically synthesized GSS was applied to ovarian fragments prepared from adult female *P. pectinifera* during the breeding season, the mature oocytes underwent germinal vesicle breakdown (GVBD) and were released from the ovary after approximately 30 min [19,21]. On the other hand, no spawning activity was observed in the case of the A chain or the B chain alone or in the mixed solution of the A and B chain. This strongly suggests that a heterodimeric structure linkaged with disulfide bonds is important for the physiological activity of GSS. Additionally, after injection with GSS into female (Figure 1C) or male of *P. pectinifera* (Figure 1D), shedding of mature eggs or spermatozoa from the gonopores was observed after about 30 min. These results strongly suggest that the GSS identified is the natural GTH-like active substance of *P. pectinifera*.

The A chain of GSS harbors a cysteine motif (Cys-Cys-X-X-X-Cys-X-X-X-X-X-X-X-X-Cys) similar to that of a member of insulin superfamily (Figure 2A). Thus, GSS is a member of the insulin superfamily. The insulin superfamily is divided into distinct two groups, one for the insulin/insulin-like growth factor (IGF) subfamily and another for the relaxin/insulin-like peptide (ILP) subfamily. A phylogenetic tree of the insulin superfamily suggested that GSS belongs to the relaxin/ILP subfamily [19]. To adequately emphasize the biochemical characteristics of this peptide hormone, starfish gonadotropic hormone was re-designated as a relaxin-like gonad-stimulating peptide (RGP) instead of a GSS [18,20,22].

## 3. Orthologs of RGP

The chemical structures of RGP have been identified in several species of starfish, *Asterias amurensis* (Aam-RGP) [23], *Asterias rubens* (Aru-RGP) [15,24,25], *Aphelasterias japonica* (Aja-RGP) [26], *Astropecten scoparius* (Asc-RGP) [27,28], and the crown-of-thorns starfish (COTS) *Achanthaster* cf. *solaris* (Aso-RGP) [29,30,31,32] (Figure 2A). Spawning-inducing activities were recognized in chemically synthesized Ppe-RGP, Aam-RGP, Aru-RGP, Aja-RGP, Asc-RGP, and Aso-RGP [19,23,25,26,27,28,32]. The class Asteroida is classified into seven orders: Forcipulatida, Paxillosida, Velatida, Valvatida, Spinulosida, Notomyotida, and Brisingida. Based on the Ppe-RGP cDNA sequence (GenBank: AB496611), RGP homologs were found by searching the sequence databases. Putative RGP precursors were identified by BLAST analysis against Trinity-assembled (https://github.com/trinityrnaseq, accessed on 8 February 2020) contig sequences with the genomic and transcriptome sequence data from more than twenty species of starfish belonging to the orders Forcipulatida, Paxillosida, Velatida, Valvatida, and Spinulosida and excluding those of the orders Notomyotida and Brisingida. Peptide sequences of identified and putative RGP precursors in twenty-four species are shown in Figure 3A. As already described in Section 2, the precursor peptide structures are arranged with signal peptides at the N-terminus, followed by B chains and C-peptides and A chains at the C-terminus.

Molecular structures of twenty-four RGP orthologs after eliminating the signal and C-peptide from the precursor peptides are shown in Figure 2A. The cysteine positions and disulfide bonds in the A and B chains are completely identical among twenty-four RGP orthologs (Figure 2A). When the amino acid types are color-coded according to their properties, most of the amino acid residues comprising RGP are well conserved for each order in species which they belong.

In the order Forcipulatida, the positions of basic amino acids Lys, Arg, and His in blue, acidic amino acids Asp and Glu in red, and Gly in light blue are completely consistent in seven species of RGP orthologs, Aam-RGP, Aru-RGP, Aja-RGP, *Marthaserias glacialis* RGP (Mgl-RGP), *Pisaster ochraceus* RGP (Poc-RGP), *Coscinasterias acutispina* (Cac-RGP), and *Labidiaster annulatus* (Lan-RGP). In the order Valvatida, the position of Gly is quite consistent in eight RGP orthologs, Ppe-RGP, Aso-RGP, *Patiria miniate* RGP (Pmi-RGP), *Parvulastra exigua* RGP (Pex-RGP), *Paterella regularius* RGP (Pre-RGP), *Asteropsis carnifera* RGP (Aca-RGP), *Certordoa semiregularis* RGP (Cse-RGP), and *Glabraster antarctica* RGP (Gan-RGP).

Interestingly, the C-terminal residues of the B chain in RGPs of the order Paxillosida (the family Astropectinidae), Asc-RGP, *A. latespinosus* RGP (Ala-RGP), and *A. duplicatus* RGP (Adu-RGP), and the order Spinulosida, *Henricia leviuscula* RGP (Hle-RGP) and *Echinaster spinulosa* RGP (Esp-RGP), are Gly-Arg, which is a potential substrate for the formation of an amidated C-terminus (Figure 2A). It was revealed that the C-terminal side of the B chain in natural Asc-RGP is amidated [28]. Thus, presumably, putative Ala-RGP, Adu-RGP, Hle-RGP, and Esp-RGP are amidated at the C-terminus of the B chain (Figure 2A).

In contrast, amidation signals are not found at the C-terminal of the B chain in *Luidia quinarian* RGP (Lqu-RGP), *L. clathrate* (Lcl-RGP), and *L. senegalensis* (Lse-RGP) of the order Paxillosida, the family Luidiidae (Figure 3A). This suggests that the C-terminus of the B chain in Lqu-RGP, Lcl-RGP, and Lse-RGP is not amidated.

On the other hand, it was revealed that chemically synthesized Asc-RGPs with a non-amidated and amidated C-terminus in the B chain exhibited equivalent induction of oocyte maturation and ovulation in an ovarian fragment of *A. scoparius* [27,28]. This suggests that C-terminal amidation in the B chain does not affect gonadotropic action in Asc-RGP. Because C-terminal amidation confers protease resistance to the peptide, RGP orthologs with amidation at the B chain are expected to be resistant against proteolytic enzymes.

Furthermore, the N-terminal amino acid residues of the B chains in Hle-RGP, Esp-RGP, Lqu-RGP, Lcl-RGP, and Lse-RGP are Gln (Figure 3A). Thus, probably, the amino acid residues at the N-terminus of these RGP orthologs are pyroglutamic acid (Pyr, pQ) (Figure 2A).

RGP and ISL are present not only in starfish (class Asteroidea), but also in other echinoderms such as sea urchins (class Echinoidea) [33], sea cucumbers (class Holothuroidae) [34,35], brittle stars (class Ohiuroidea) [36], and sea lilies (class Crinoidea) [37]. It was revealed that sea cucumber RGP induces gamete shedding in *Holothuria scabra* and *H. leucospilota* [35]. On the other hand, it is also known that cubifrin induces oocyte maturation and spawning behavior in sea cucumber *Apostichopus japonicus* [38]. Cubifrin, unlike RGP, is a short C-terminal amidated peptide (NGIWY amide) [38]. It has also been demonstrated that short C-terminal amidated neuropeptides (W/RPRP/A amide) induce oocyte maturation and spawning in jellyfish (*Clytia hemisphaerica* and *Cladonema pacificum*) [39]. The neuropeptides are secreted from nervous cells in the ovarian epithelium by light stimulation. Thus, the jellyfish neuropeptide is a MIH rather than a GTH-like peptide. In bivalves, serotonin as an MIH induces oocyte maturation and ovulation [40]. Ascidians are evolutionarily located between echinoderms and chordates. A vasopressin-like peptide secreted from the neural complex induces oocyte maturation and ovulation in the cosmopolitan ascidian *Ciona intestinalis* [41]. A cholecystokinin/gastrin homolog, cionin, induces ovulation in *C. intestinalis* [42]. Thus, various neuropeptides are responsible for triggering oocyte maturation and ovulation in invertebrates.

Furthermore, genome and transcriptome analysis in ambulacrarians revealed two relaxin-type peptides, RGP and a second relaxin-like peptide 2 (RLP2), in *A. rubens* (Aru-RLP2), *M. glacialis* (Mgl-RLP2), *P. pectinifera* (Ppe-RLP2), *P. miniate* (Pmi-RLP2), and COTS (Aso-RLP2) (Figure 2B and Figure 3B) [24,35,43]. It was recognized that Aru-RLP2 and Ppe-RLP2 induced oocyte maturation and ovulation in ovarian fragments (unpublished data). This strongly suggests that RLP2 has GTH-like activity. Thus, RGP and RLN2 should be renamed RGP1 and RGP2, respectively. On the other hand, it is unclear how RGP and RLP2 are physiologically related in oocyte maturation and ovulation.

## 4. Phylogenetic Analysis of RGP

A molecular phylogenetic tree was constructed using the amino acid sequences of twenty-one RGP precursors (Figure 4A). Because the precursor of Lse-RGP is a partial sequence, the phylogenetic tree was constructed with 23 RGP orthologs, excluding Lse-RGP. The orders Valvatida, Forcipulatida, Paxillosida, Spinulosida, and Velatida are indicated by purple, yellow, light-green (the family Astropectinidae), green (the family Luidiidae), orange, and gray backgrounds. These RGP sequences form separate clusters for each order, except for the order Paxillosida (Figure 4A).

It had been considered that starfish species of the order Paxillosida are a primitive group, as judged by their inability to evert the cardiac stomach and the lack of the brachiolaria stage in their larval development [44]. Mah and Folts (2011) [44] reported in living taxa among the Neoasteroidea that the Asteroidea is divided into two superorders: one for Valvatacea, including the orders Valvatida and Paxillosida, and another for Forcipulatacea, including the order Forcipulatida. The family Luidiidae was formerly classified independently as the order Platyasterida [45]. Today, the order of Platyasterida is invalid [46]. The characteristics of RGP molecules are different between the family Astropectinidae and the family Luidiidae (Figure 4A). If a lot of molecular and morphological differences are found between Astropectinidae and Luidiidae, it may be possible that the order Platyasterida will come back for Asteroidea.

Phylogenetic analyses of RGP and RLP2 were also examined for two species from the order Forcipulatida (*A. rubens* and *M. glaciaris*) and three from the order Valvatida (*P. pectinifera*, *P. miniate*, and *A*. cf. *solaris*). RGP and RLP2 groups formed separate clusters (Figure 4B). This suggests that RGP and RLP2 are paralogs. It seems likely that these peptides evolved independently in the class Asteroidea.

## 5. Localization of RGP

Repeatedly, GSS/RGP has been shown to be mainly present in the radial nerve cords and circumoral nerve rings in *A. amurensis* and *P. pectinifera* [14,47] (Figure 5A). A bioassay using ovarian fragments also showed the presence of low concentrations of GSS/RGP in the tube foot, body wall, and cardiac stomach, but no GSS/RGP activity was detected in the pyloric caeca, ovaries, or testes [47]. Immunoassays using specific antibodies of Ppe-RGP [48] and Aam-RGP/Aru-RGP [49] indicated that high concentrations of RGP exist in the radial nerve cords of *P. pectinifera* [50,51] and *A. rubens* [52] but are not detectable in the cardiac stomach, pyloric stomach, pyloric caeca, tube feet, ovaries, or testes in *P. pectinfera* [50,51] or *A. rubens* [52]. Furthermore, the real-time quantitative polymerase chain reaction (RT-qPCR) method showed the presence of RGP precursor transcripts at high levels in the radial nerve cords and at lower levels in the cardiac stomach and pyloric caeca, with trace levels in the tube feet, ovaries, and testes [20].

More specific localization of RGP expression in starfish at a cellular level has been achieved using mRNA in situ hybridization. The expression of RGP mRNA was observed in the ectoneural epithelial layer in the radial nerve cords of *P. pectinifera* [19]. In *A. rubens*, cells expressing RGP transcripts were revealed in the ectoneural epithelium of the radial nerve cords, the circumoral nerve ring, and the tube feet [25]. Furthermore, an immunohistochemical analysis using RGP-specific antibodies revealed that RGP is distributed not only in the epithelium of the ectoneural region but also in the neuropile of the ectoneural region in the radial nerve cords of crown-of-thorns starfish (COTS) [32]. This suggests that RGP synthesized in the epithelial cells around the radial nerve cords is transported to the ectoneural neuropile regions and accumulates there.

From the histological study [53], it seems that GSS/RGP is transported from the radial nerve cords to the radial haemal sinus. This sinus is connected to axial, visceral, and genital haemal sinuses of the haemal system [54]. Thus, it is likely that RGP is transported from the radial nerve cords to the gonads through the haemal system. On the other hand, it has been hypothesized that GSS/RGP secreted from the radial nerve cords acts on ovarian follicle cells via coelomic fluid [14]. It has been demonstrated that a calcium ionophore, ionomycin, induces the release of RGP from radial nerve cords [55]. This suggests that an increase in intracellular calcium ions is involved in RGP secretion. However, the gonadotropin-releasing hormone-like peptide had no effect on RGP secretion from radial nerve cords in *P. pectiniera* [22]. The neuropeptide SALMFamide also inhibited GSS/RGP secretion from radial nerve cords [56]. Probably, unknown neuropeptides or neurotransmitters are involved in the RGP secretion.

On the other hand, an extensive population of cells expressing RGP transcripts was observed in the arm tips of *A. rubens* (Figure 5A), largely concentrated in the body wall external epithelium surrounding the sensory terminal tentacle and the associated optic cushion (simple eye) [25]. Thus, it is possible that the arm tips, and not the radial nerve cords, may be the physiological source of RGP that triggers spawning in starfish. The rationale for this hypothesis is that RGP-expressing cells in the arm tip are located proximal to the terminal tentacle, and the associated sensory organs are ideally positioned to integrate sensation of changes in the environmental conditions (for example, day length, lunar cycle, or pheromones) thought to trigger spawning. However, it is still unknown whether the RGP-expressing cells in the arm tips have axonal processes that terminate at sites whereby RGP secreted by these cells can gain access to the coelomic cavity where the gonads are located.

Recently, it was revealed in gravid COTS that RGP is significantly upregulated in both male and female coelomocytes compared to in other tissues/organs [57] (Figure 5A). Thus, possibly, RGP stored in coelomocytes plays some roles in reproduction.

Furthermore, it was revealed that RGP is expressed in the gonoducts of *A. rubens* proximal to its gonadotropic site of action in the gonads [58] (Figure 5A). This finding is important because it provides a new perspective on how RGP may act as a gonadotropin in starfish. Thus, it is highly possible that the secretion of RGP from the gonoducts triggers gamete maturation and spawning in starfish because, prior to spawning, the RGP-expressing neurons in the gonoducts secrete RGP to diffuse as a neurohormone into the lumen of the gonoduct and then into the lumen of the gonads (Figure 5A).

## 6. Hormonal Action of RGP

Although injection of RGP in female and male starfish brings about gamete shedding (Figure 1C,D), the hormonal action is indirect. RGP stimulates ovarian follicle cells [59,60] (Figure 1A) and testicular interstitial cells [61] (Figure 1B) to produce 1-MeAde. RGP binds specifically to a membrane preparation of follicle cells and interstitial cells [62,63]. When follicle cells in *P. pectinifera* or *A. scoparius* are incubated with Ppe-RGP or Asc-RGP, a dose-related increase in cyclic AMP (cAMP) production is observed, coinciding with an increase in 1-MeAde production [19,27,63,64,65].

A potent inhibitor of phosphodiesterase, 3-isobutyl-1-methylxanthine (IBMX), also induces both cAMP and 1-MeAde production in *P. pectinifera* follicle cells [65,66,67,68,69]. Although GSS/RGP slightly stimulates adenylyl cyclase activity in the crude membrane preparation of *P. pectinifera* follicle cells, GTP markedly enhances this action of GSS/RGP in a dose-dependent manner [70]. Nonhydrolyzable GTP analogs also activate adenylyl cyclase activity. These suggest that a stimulatory type of guanine nucleotide regulatory binding protein (Gs-protein) and adenylyl cyclase are involved in the action of GSS/RGP on 1-MeAde production of follicle cells. Therefore, cAMP plays an important role in mediating RGP action on 1-MeAde production. It is possible that the receptor of RGP (RGPR) is a G protein-coupled receptor (GPCR). The action of RGP is mediated through the activation of its receptor, Gs-protein, and adenylyl cyclase in the follicle cells of *P. pectinifera* [19,63,70] (Figure 6B).

On the other hand, RGP fails to induce 1-MeAde and cAMP production in follicle cells of *P. pectinifera* ovaries during oogenesis. By the maturation stage, thus, follicle cells must acquire the ability to respond to RGP to produce 1-MeAde and cAMP [71]. In contrast, the specific binding of RGP to membrane preparations of follicle cells is detectable regardless of oogenesis or maturation stage in *P. pectinifera* [72,73,74]. This suggests that RGP receptors are present in follicle cells at the oogenesis stage. According to Western blotting analyses using anti-Gαs antibodies, Gαs is deficient in oogenesis-stage follicle cells in *P. pectinifera* [72,73]. The activity of Gαs transcript in *P. pectinifera* follicle cells increases significantly as the size of the oocytes increases, whereas the mRNA level of Gαi is almost constant regardless of oocyte size [73]. Thus, de novo synthesis of the Gαs protein is necessary for follicle cells to respond to RGP for 1-MeAde and cAMP production (Figure 6B).

## 7. RGP Receptor

From a binding assay using [^125^I]labeled Ppe-RGP, it was revealed that Ppe-RGP significantly interacts with the cell membrane preparations of ovarian follicle cells in *P. pectinifera* [62,71,72]. This strongly suggests that RGP receptor (RGPR) exists in the surface of ovarian follicle cells. Ppe-RGPR was identified by BLAST analysis against Trinity-assembled (https://github.com/trinityrnaseq, accessed on 25 April 2023) contig sequences with the transcriptome sequence data of follicle cells in *P. pectinifera* (SRR8627925 and SRR8627926) using human relaxin family receptors RXFP1_human (Q9HBX9) and RXFP2_human (Q8WXD0) as queries. Two contig sequences were obtained as putative Ppe-RGP receptors, Ppe-RGPR1 (Figure 7A) and Ppe-RGPR2 (Figure 7B) [75]. The ORFs of putative Ppe-RGPR1 and Ppe-RGPR2 consist of 2538 bp and 2844 bp and encode 845 and 947 amino acids, respectively. Both putative Ppe-RGPR1 and Ppe-RGPR2 are GPCRs harboring seven transmembrane (TM) domains [72]. There are leucine-rich repeat (LRR) domains in the extracellular region of the N-terminal segment. Putative Ppe-RGPR1 bears one low-density lipoprotein receptor class A (LDLa), and Ppe-RGPR2 has two LDLa domains. There are seven LRR domains in RGPR1 (Figure 7A) and nine in RGPR2 (Figure 7B), respectively. A calcium-binding motif [Asp/Asn-X-X-X-Asp-X-X-Asp/Asn-X-X-Asp-Glu] [76] is also found in LDLa domain (Figure 7A,B). The predicted three-dimensional (3D) structure of Ppe-RGPR2 is shown in Figure 7C. These structural features are reminiscent of vertebrate RXFPs. Thus, Ppe-RGPR1 and Ppe-RGPR2 belong to subgroup δ of the rhodopsin class (Type I or Class A) of GPCRs and are further classified as subtype C LGRs [77]. Orthologs of Ppe-RGPR1 or Ppe-RGPR2 have also been found in transcriptome data of *A. rubens* and COTS, respectively [43,57].

Sf9 cells expressing either Ppe-RGPR1 or PpeRGPR2 respond to Ppe-RGP in a dose-dependent manner, but not to Aam-RGP [75]. This is in accordance with the fact that Ppe-RGP induces 1-MeAde production in *P. pectinifera* follicle cells, but Aam-RGP does not [19]. The responses in Ppe-RGPR2 are statistically significant [75]. Thus, presumably, Ppe-RGPR2 is a natural candidate for Ppe-RGPR. On the other hand, Ppe-RGPR1 is an ortholog to hRXFP1 and hRXFP2. Thus, it is possible that Ppe-RGPR1 is also a receptor for Ppe-RGP. Presumably, RGPR2 is a paralog to RGPR1.

In humans, the relaxin (RLN)-like peptide family consists of seven peptides, RLN1, RLN2, RLN3, insulin-like (INSL) peptide 3, INSL4, INSL5, and INSL6 [78]. It is also known that RXFPs belong to the superfamily of rhodopsin-like GPCRs [79]. The receptors for each peptide are shown as follows: RLN2, INSL3, RLN3, and INSL5 signal through RXFP1, RXFP2, RXFP3, and RXFP4, respectively [80,81,82,83,84,85,86]. RNL1 is possibly a ligand of RXFP1. However, the native receptors for INSL4 and INSL6 are yet to be identified [81].

It has been revealed that the B chain of RLN and related peptides plays an important role in binding to the receptor [83,86,87]. Despite its similarity to the RLN superfamily, the RGP sequence does not possess the vertebrate ‘relaxin-specific receptor-binding cassette’ (Arg-X-X-X-Arg-X-X-Ile/Val) which is a distinct and well-conserved feature of the RLN group B chains [88,89]. A comparison of the amino acid sequences of the middle region of the B chains indicated that residues of the ‘receptor-binding cassette’ correspond to Asp^B6^, Met^B10^, and Phe^B13^ for Ppe-RGP and Glu^B7^, Met^B11^, and Tyr^B14^ for Aru-RGP and Aja-RGP (Figure 6A). The 3D structure models of the B chain appear to be quite similar for Ppe-RGP and Aru-RGP [90,91] (Figure 6A). Therefore, Asp/Glu-X-X-X-Met-X-X-Phe/Tyr in the B chain of RGPs may be involved in binding to its cognate receptors, similar to in vertebrates.

In *Drosophila melanogaster*, eight insulin-like peptides (Dilp 1-8) are encoded on separate genes. Dilp8, which was discovered recently, acts on relaxin receptor homolog Lgr3, which is a leucine-rich repeat-containing GPCR [92]. DilP8 is expressed and present in follicle cells around the mature eggs of *D. melanogaster* [9]. Furthermore, Lgr3 expression has been detected in follicle cells close to the oviduct, in addition to in numerous neurons of the brain and abdominal heart [9]. Thus, presumably, DilP8 plays an important role in reproduction, particularly ovulation, in *D. melanogaster*. From phylogenetical analysis, it was revealed that Lgr3 in *D. melanogaster* is an ortholog to hRXFP1 and hRXFP2 [92].

## 8. Species Specificity of RGP

Generally, RGP (GSS) acts in a non-species-specific manner, with some exceptions [93,94,95]. Ppe-RGP can induce oocyte maturation and ovulation in ovarian fragments of *A.* cf. *solaris* [29,32], *A. amurensis* [23], *A. japonica* [26], *A. rubens* [25], and *A. scoparius* [27,28]. Although Aam-RGP induces spawning in the *A. japonica* ovary, Aam-RGP and Aja-RGP are inactive in the *P. pectinifera* ovary. The mechanism of species specificity in RGP was examined using chimeric RGP derivatives comprising different combinations of A and B chains in Ppe-RGP, Aru-RGP, and Aja-RGP [90,91]. As a result, it was revealed that the B chain of RGPs, particularly Asp/Glu-X-X-X-Met-X-X-Phe/Tyr, plays an important role in binding to the receptor. On the other hand, the Pro^A17^ of Ppe-RGP and Arg^A18^ of Aru-RGP and Aja-RGP are located near the B chain (Figure 6A). The side chain of arginine is positively charged and is larger than the side chain of proline. Thus, it seems likely that the side chain of the amino acid constituting the A chains of Aru-RGP and Aja-RGP impairs binding to the Ppe-RGP receptor.

In contrast, Aso-RGP induces spawning in *P. pectinifera* ovaries [32]. Amino acid sequences of the middle region of B chains in Aso-RGP are Asn^B6^, Leu^B10^, and Tyr^B13^. The 3D structure models of Ppe-RGP and Aso-RGP reveal that key residues are located in the same helical turn in the B chain (Figure 6A). Additionally, Pro^A17^ in the A chain of Aso-RGP, which may affect steric hindrance with the Ppe-RGP receptor, is the same as that of Ppe-RGP. Thus, it is suggested that Ppe-RGP and Aso-RGP are capable of binding to both receptors.

## 9. RGP-Induced 1-MeAde Production

Upon application of RGP to ovarian follicle cells, 1-MeAde is produced immediately [64,96]. Within a few minutes of RGP treatment, the concentration of 1-MeAde in the ovaries reaches a sufficient level to induce oocyte maturation [97]. However, 1-MeAde is absent in follicle cells before RGP treatment [98]. This suggests that 1-MeAde produced under the influence of RGP is newly synthesized (Figure 6B), rather than previously stored within follicle cells or a breakdown product of some 1-MeAde-containing substance, such as ribonucleic acid [99].

Regarding 1-MeAde biosynthesis, methionine and S-adenosylmethionine (SAM) were found to enhance 1-MeAde production in GSS/RGP-stimulated follicle cells [100,101] (Figure 6C). In addition, the [^14^C]-methyl group of methionine converts to 1-MeAde in follicle cells stimulated by GSS/RGP [100,101,102,103]. This suggests that the biochemical role of RGP in 1-MeAde production is activation of the transfer of the methyl group to the N^1^ site of the purine nucleus of the 1-MeAde precursor (Figure 6C). It is possible that cAMP-dependent protein kinase (PKA) is involved in the transmethylation [104].

Furthermore, it was revealed that GSS/RGP causes a reduction in the intracellular level of ATP without any change in the levels of ADP and AMP following 1-MeAde production in *P. pectinifera* follicle cells [105,106]. Application of GSS/RGP to follicle cells that had previously accumulated radiolabeled ATP using radiolabeled [^14^C]adenine or [^14^C]adenosine produced radiolabeled 1-MeAde [106]. This suggests that the adenine moiety of 1-MeAde is derived from ATP [106]. Furthermore, when follicle cells were preincubated in [^3^H]-CH_3_-methionine and [^14^C] adenine, it was revealed that one mol of 1-MeAde was synthesized in follicle cells from one mol of ATP and one mol of methionine following GSS/RGP application [106]. Because SAM is generally synthesized from ATP and methionine by L-methionine adenosyltransferase (MAT), it seems likely that 1-MeAde is synthesized from intracellular SAM as a substrate (Figure 6C).

On the other hand, follicle cells in *P. pectinifera* can produce 1-MeAde without GSS/RGP when they are incubated with 1-methyladenosine (1-MeAdo) [107] and 1-methyladenosine monophosphate (1-MeAMP) [108]. The enzyme 1-MeAdo ribohydrolase exists in follicle cells [109,110]. This suggests that 1-MeAdo and 1-MeAMP are intermediates in 1-MeAde biosynthesis. In addition, 1-MeAde is synthesized non-enzymatically by heat treatment of SAM [111]. It is possible that RGP activates the reaction from SAM to 1-MeAdo or 1-MeAMP. It might be involved in an unknown enzyme, SAM transmethylase (Figure 6C).

## 10. Gamete Shedding

Although the contraction of the gonadal walls is essential for shedding eggs and spermatozoa from the narrow gonopores, RGP and 1-MeAde do not have the physiological effects of contracting the ovaries and testes [112]. This suggests that a contraction-inducing substance is present within starfish gonads. 1-MeAde is also produced by testicular somatic interstitial cells under the influence of GSS/RGP [61] (Figure 1B). Unlike immature oocytes in the ovary, starfish spermatozoa have already accomplished meiotic division in the testis. Thus, the role of 1-MeAde in males remains unknown.

In the case of sea urchins, the shedding of eggs and spermatozoa can be easily induced by injecting isotonic potassium chloride or acetylcholine (ACh) into the body cavity. In contrast, even if ACh is injected into a mature starfish, spawning never happens. Despite this, it has been revealed that ACh exists in the ovaries and testes of starfish [113]. Through analysis using tandem imaging mass spectrometry, ACh was detected in *P. pectinifera* ovaries and testes, particularly in oocytes and follicle cells in the ovaries and spermatozoa in the testes [113]. This suggests that ACh is stored in the oocytes and follicle cells in the ovaries. On the other hand, since several kinds of somatic cells, such as amoeboid and interstitial cells, are contaminated in the lumen of starfish testes [114,115,116,117,118], ACh is thought to exist in somatic cells other than spermatozoa in testes. In contrast, ACh is hardly observed in the peripheral region of the gonads where the nervous system should be distributed, suggesting that ACh in ovaries and testes is independent of the nervous system. In general, non-neuronal ACh is predicted to function in auto- and paracrine manners, which cover the local regulatory actions of ACh in cells or trophic molecules [119,120,121]. Because non-neuronal ACh is also substantially involved in the regulation of reproduction in vertebrates and invertebrates [121], it is possible that ACh in the lumen of gonads is involved in the contraction of gonadal walls in starfish. In addition, ACh is present in unfertilized eggs of sea urchins and plays an important role during early development [122,123]. It may be possible that ACh plays a similar role in starfish oocytes.

Shirai and coworkers reported previously that the molecular weight of a contraction-inducing substance contained 1-MeAde-treated egg jelly is predicted to be more than 25,000 Da [112]. This is much larger than 146.2, the molecular weight of ACh. On the other hand, it has been reported that ACh-binding protein is present in snails [124,125]. Although it is unclear whether an ACh-binding protein is present in starfish, it may be possible that ACh secreted from oocytes or follicle cells binds to the binding proteins with the increase in 1-MeAde concentration. This leads to the hypothesis that 1-MeAde plays another role in the ovaries to promote the secretion of ACh other than the induction of oocyte maturation. It is also possible that 1-MeAde induces the secretion of ACh in testes to shed spermatozoa (Figure 1A,B).

Furthermore, Ca^2+^ influx was shown to be important for gonadal contraction in starfish because a potent inhibitor of voltage-dependent calcium channel (VDCC), verapamil, inhibited 1-MeAde-induced spawning in *P. pectinifera* [113]. The muscarinic ACh receptor antagonist atropine also inhibited 1-MeAde-induced ovulation. Presumably, ACh activates muscarinic ACh receptors to stimulate Ca^2+^ influx via the VDCC in gonadal walls (Figure 5B).

Therefore, during breeding season, RGP secreted from the ectoneural neuropile regions in radial nerve cords [32], arm tips [25], coelomocytes [57], and/or gonoducts [58] is transported to the gonads. RGP acts on ovarian follicle cells and testicular interstitial cells to produce 1-MeAde [59,61] (Figure 1A,B). In ovaries, 1-MeAde stimulates oocytes to produce MPF, which is a direct trigger for GVBD and FEBD [126]. After the FEs separate from the oocytes, the denuded mature oocytes become free within the ovary [97,112,127]. Simultaneously, 1-MeAde induces ACh secretion within the ovaries and testes. ACh acts on muscarinic ACh receptors on muscle cells in the gonadal walls. Finally, ACh brings about peristaltic contractions caused by Ca^2+^ influx via VDCC in the gonads in order to shed gametes [113] (Figure 5B).

Furthermore, l-glutamic acid was identified as a spawning inhibitor that antagonizes the action of RGP in the gonads of *P. pectinifera* [128,129]. Application of ACh to *P. pectinifera* ovaries, under the conditions that 1-MeAde-induced spawning was inhibited by l-glutamic acid, restored shedding of the mature eggs from ovaries [130]. Thus, it may be possible that l-glutamic acid prevents ACh secretion from shedding gametes during oogenesis and spermatogenesis.

## 11. RGP in Larvae

In adult starfish, RGP is mainly distributed in the nervous system, such as in the radial nerve cords and circumoral nerve rings [25,29,50,51]. It has also been demonstrated that nervous systems are developed along ciliary bands of starfish embryos and larvae before metamorphosis [131,132]. Thus, it is possible that RGP is expressed in the nervous system of embryos and larvae. In fact, spawning-inducing activities were observed in extracts from embryos and larvae of *P. pectinifera* [133]. The spawning-inducing activity was due to RGP, not 1-MeAde. The transcript activity of Ppe-RGP was also found in brachiolaria. These findings suggest that Ppe-RGP is synthesized in the larval stage during the early development of *P. pectinifera*. From an immunohistochemical analysis using specific antibodies for Ppe-RGP, it was seen that Ppe-RGP is distributed in the peripheral adhesive papilla of the brachiolaria arms in *P. pectinifera* but not in the adult rudiment and ciliary band regions [133] (Figure 8). It may be possible that RGP is newly synthesized in the radial nerve cords and circumoral nerve rings after metamorphosis, although RGP exists in the larvae before metamorphosis.

On the other hand, the specific binding of Ppe-RGP to the receptor is not found in the embryos and larvae of *P. pectinifera* [133]. Although RGP plays a role in inducing 1-MeAde production, 1-MeAde is not detected in embryos and larvae in *P. pectinifera*. Presumably, RGP receptor is absent in the larval stages before metamorphosis. If RGP is involved in the induction of metamorphosis from larvae to juveniles, the receptor may appear during the metamorphosis.

It is presumed that the adhesive papilla is important for adhesion to the bottom surface during metamorphosis [134]. However, exposure of Ppe-RGP to late brachiolaria larvae of *P. pectinifera* showed no effect on metamorphosis [133]. It may be possible that RGP in the adhesive papilla acts as a kind of sensor for settlement from floating to determine where to metamorphic transform. This might be a new physiological action of the relaxin superfamily.

## 12. Conclusions

The chemical structure of the gonadotropin-like active substance identified in the marine invertebrate starfish is a peptide hormone similar to that of relaxin, which plays a physiological role in pregnancy in mammals, including humans. The active substance was renamed RGP from GSS due to its chemical structure and physiological action. RGP activates receptor/G protein/adenylyl cyclase activity in ovarian follicle cells and testicular interstitial cells to induce the production of 1-MeAde. RGP receptors are GPCRs as well as the relaxin receptors. RGP indirectly induces oocyte maturation and ovulation via 1-MeAde. It is probable that 1-MeAde stimulates ACh secretion in ovaries and testes to bring about contraction of the gonad wall, resulting in the shedding of mature eggs or spermatozoa into the water column. RGP-like peptides have also been found in sea lilies, sea urchins, sea cucumbers, and brittle star. Therefore, the molecular identification of RGP as a gonadotropin in starfish has provided a basis for important advances in our knowledge and understanding of the reproductive neuroendocrinology of echinoderms. The reproductive regulation by relaxin-like peptide Dlip8 and its receptor Lgr3 has also been shown in *D. melanogaster*. Thus, relaxin signaling in the context of reproductive physiology is an evolutionarily ancient phenomenon that can be traced back to the common ancestor of deuterostomes and protostomes.

## Figures and Tables

**Figure 1 biomolecules-13-00781-f001:**
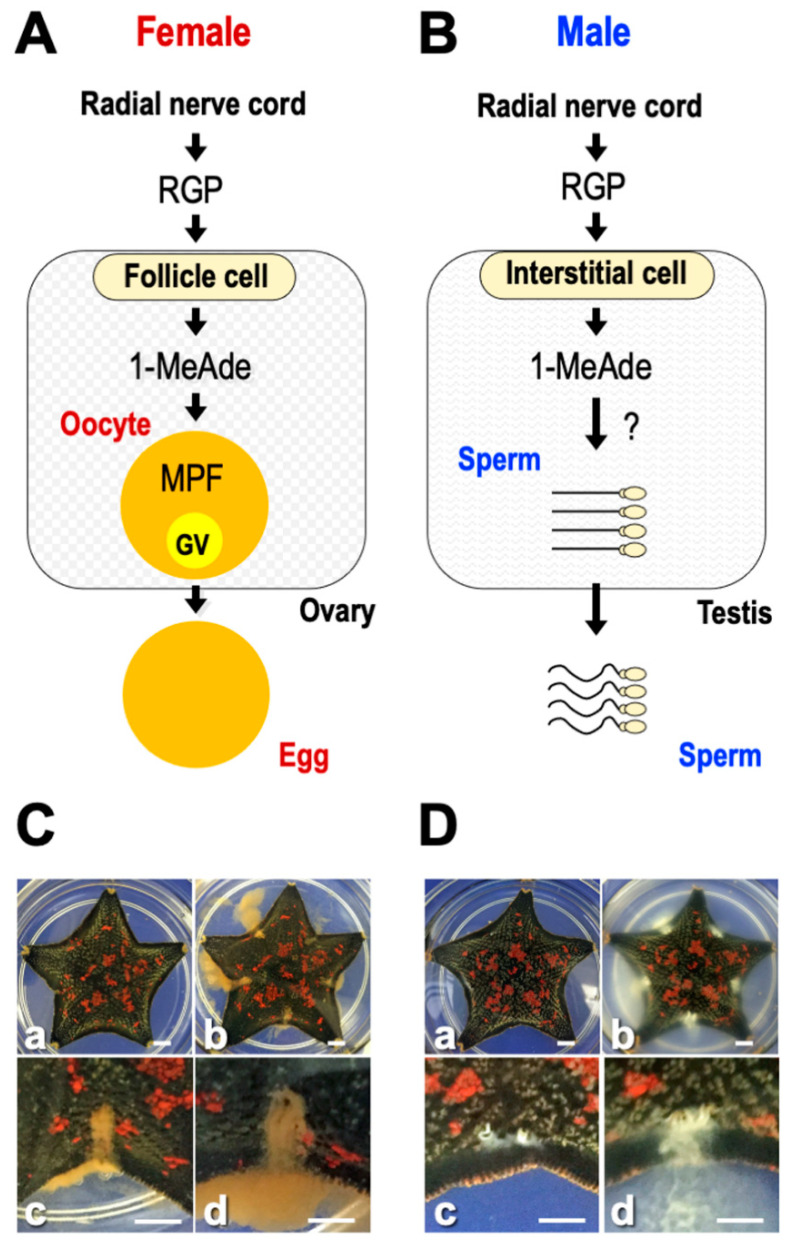
Hormonal control of relaxin-like gonad-stimulating peptide (RGP) for ovary and testis in starfish. (**A**) Regulatory mechanism of RGP on oocyte maturation and ovulation in female. (**B**) Regulatory mechanism of RGP on sperm shedding in male. (**C**) Photos of RGP-injected female shedding mature eggs. (**D**) Photos of RGP-injected male shedding spermatozoa. (**a**) Before RGP injection, (**b**) 30 min after RGP injection, (**c**) gonopores that have just begun shedding eggs or spermatozoa, and (**d**) enlarged gonopores of starfish shown in (**b**). Bars = 0.5 cm.

**Figure 2 biomolecules-13-00781-f002:**
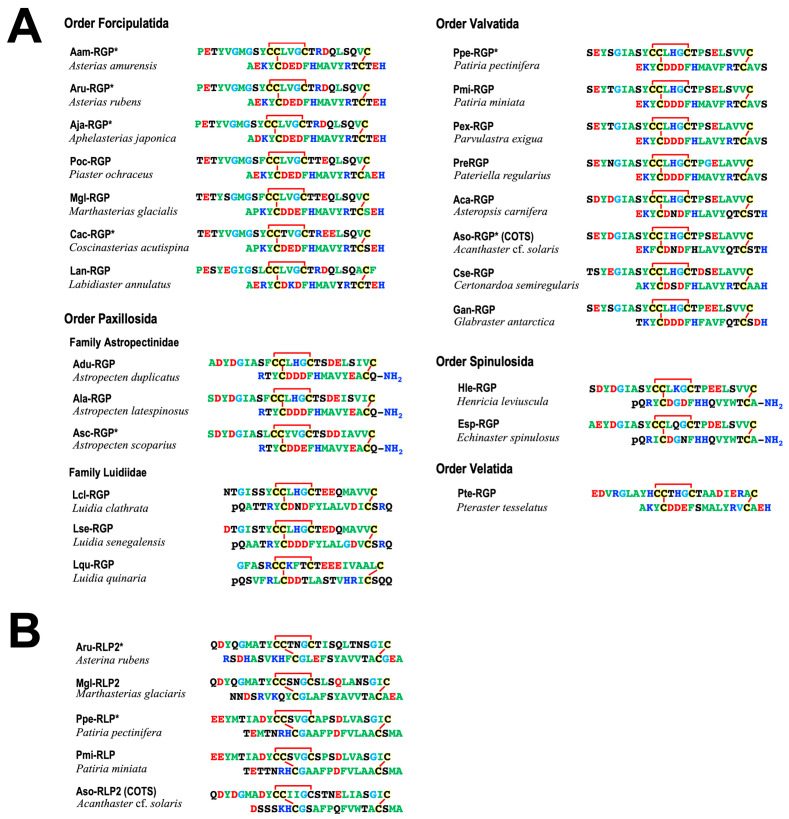
Comparison of the heterodimeric structures of relaxin-like gonad-stimulating peptide (RGP) (**A**) and relaxin-like peptide 2 (RLP2) (**B**) in various species of starfish. To illustrate the conserved features, the amino acid types are color coded according to their properties, with basic residues in blue (Arg, Lys, and His), acidic residues in red (Glu and Asp), hydrophobic residues in green (Ala, Val, Ile, Phe, Trp, Tyr, Pro, and Met), hydrophilic residues in black (Ser, Thr, Asn, and Gln), and glycine in light blue. The disulfide bridges are shown in red. *, RGPs with asterisk were chemically synthesized and confirmed the spawning-inducing activity. The peptide sequence in the upper row is the A chain, and the lower row is the B chain. Abbreviations: *Patiria pectinifera*, Ppe; *P. miniata*, Pmi; *Parvulastra exigua*, Pex; *Pateriella regularius*, Pre; *Asteropsis carnifera*, Aca; *Acanthaster* cf. *solaris*, Aso; *Certonardoa semiregularis*, Cse; *Glabaster antarctica*, Gan; *Astropecten latespinosus*, Ala; *A. scoparius*, Asc; *A. duplicatus*, Adu; *Echinaster spinulosus*, Esp; *Henricia leviuscula*, Hle; *Asterias amurensis*, Aam; *A. rubens*, Aru; *Aphelasterias japonica*, Aja; *Piaster ochraceus*, Poc; *Marthasterias glacialis*, Mgl; *Coscinasterias acutispina*, Cac; *Labidiaster annulatus*, Lan; *Luidia quinaria*, Lqu; *L. clathrata*, Lcl; and *Pteraster tesselatus*, Pte.

**Figure 3 biomolecules-13-00781-f003:**
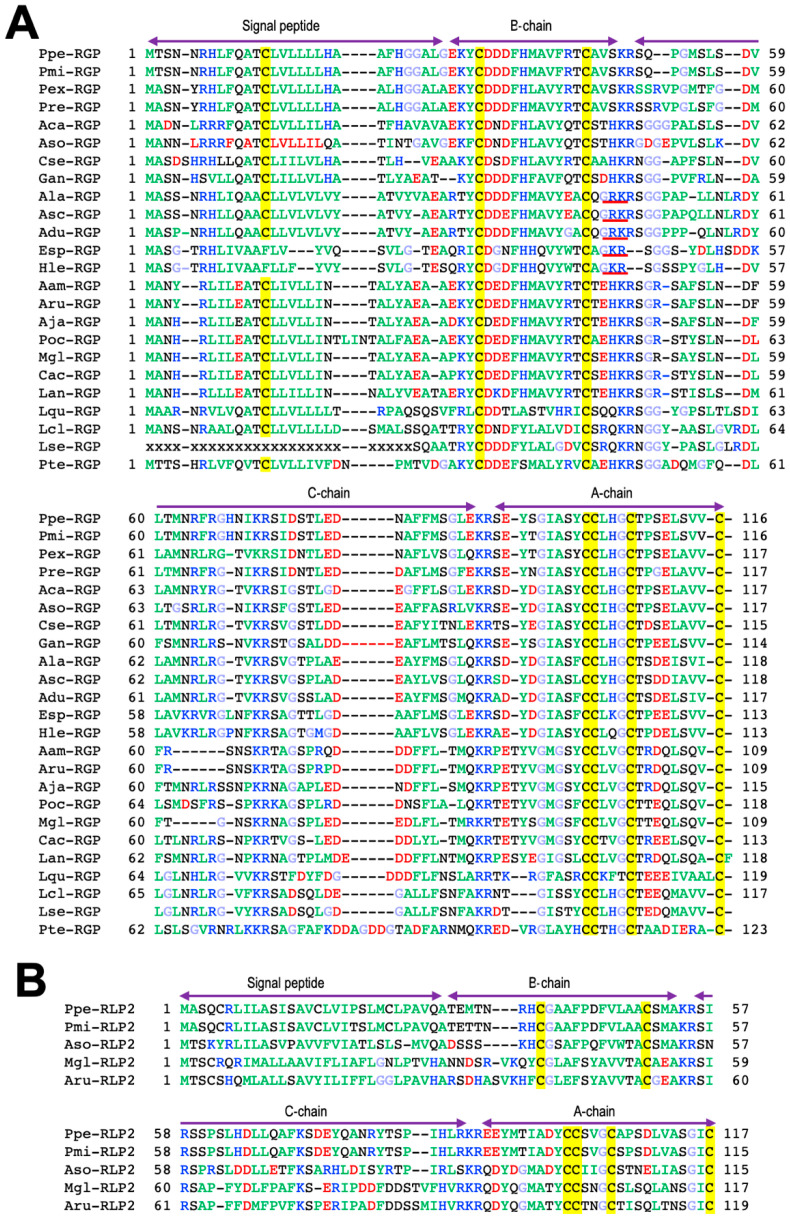
Alignment of preprohormone sequences of relaxin-like gonad-stimulating peptide (RGP) (**A**) and relaxin-like peptide 2 (RLP2) (**B**) in various species of starfish. To illustrate the conserved features, the amino acid types are color coded according to their properties, with basic residues in blue (Arg, Lys, and His), acidic residues in red (Glu and Asp), hydrophobic residues in green (Ala, Val, Ile, Phe, Trp, Tyr, Pro, and Met), hydrophilic residues in black (Ser, Thr, Asn, and Gln), and glycine in light blue. The cysteine residues are highlighted in yellow. The sequence of amide signal is shown underlined in red. Accession numbers for each RGP are listed in Appendix A. For abbreviations, refer to the caption in Figure 2.

**Figure 4 biomolecules-13-00781-f004:**
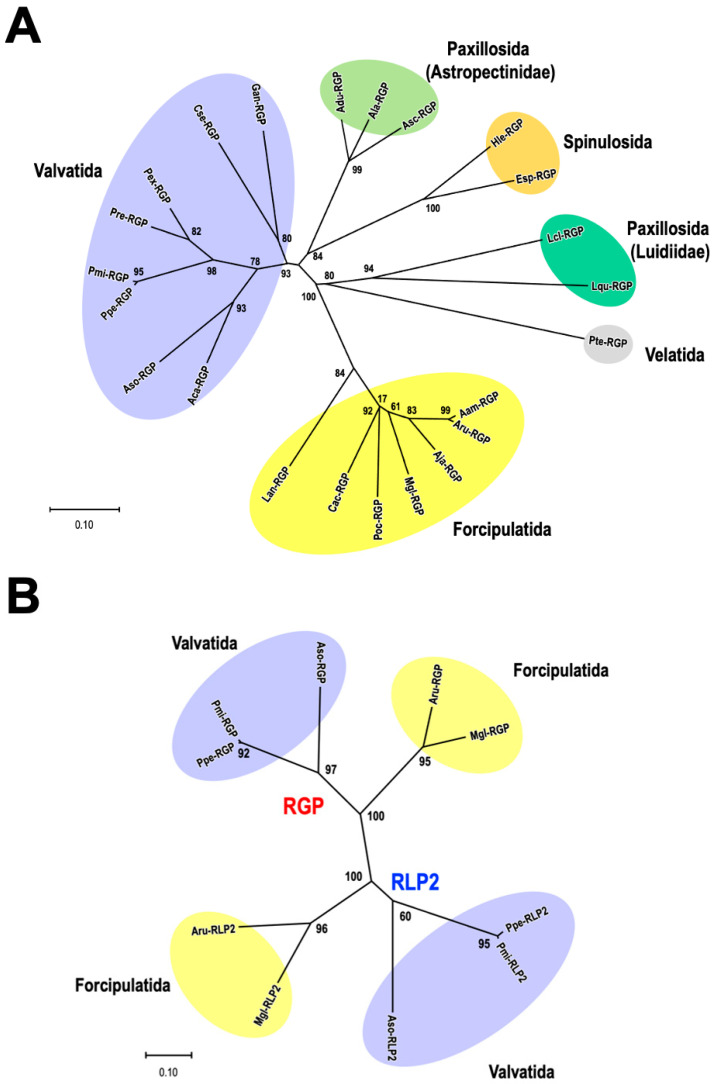
The molecular phylogeny of asteroid species based on the RGP and RLP2. (**A**) The phylogenetic tree of RGP precursor proteins in various species of starfish. Orders Valvatida, Forcipulatida, Paxillosida (the families Astropectinidae and Luidiidae), Spinulosida, and Velatida are highlighted in purple, yellow, light-green and green, orange, and gray backgrounds. (**B**) The phylogenetic tree of RGP and RLN2 precursor proteins in asteroid species belonging to the orders Forcipulatida and Valvatida. The phylogenic trees were constructed from an alignment using the neighbor-joining method with the bootstrap resampling number set at 100. The number located beside each branch is the bootstrap score. Branch lines indicate evolutionary distances. For abbreviations, refer to the caption in Figure 2.

**Figure 5 biomolecules-13-00781-f005:**
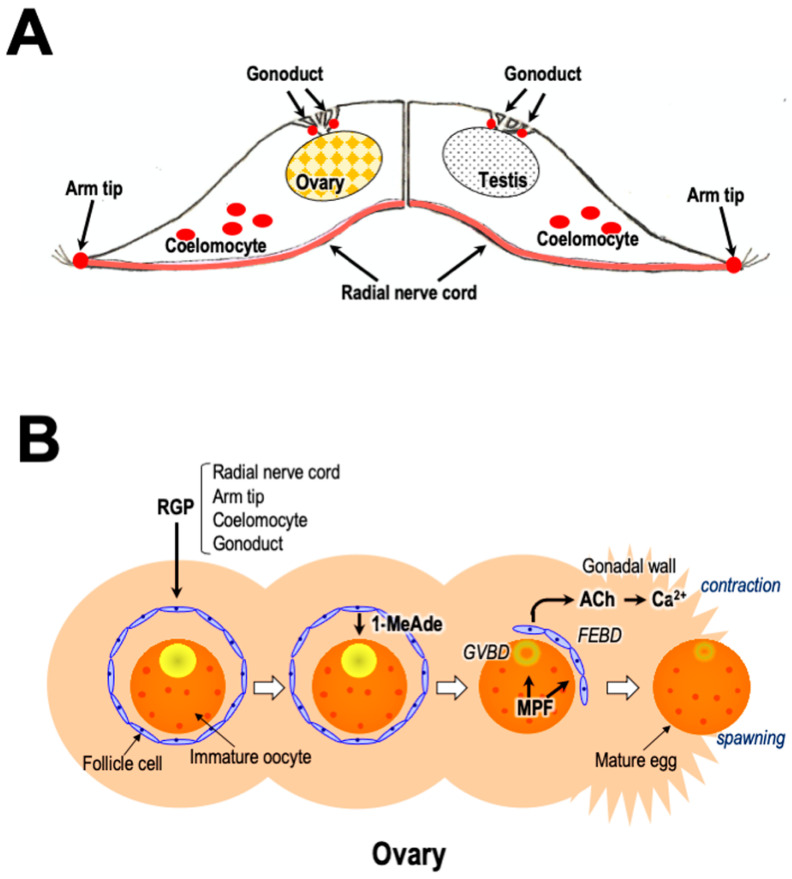
Localization of RGP and its possible regulatory mechanism for oocyte maturation and ovulation in ovaries. (**A**) RGP is mainly present in radial nerve cords but also in arm tips, coelomocytes, and gonoducts. However, RGP is not detected in ovaries and testes. (**B**) The process of RGP-induced oocyte maturation and ovulation is as follows: RGP secreted from the ectoneural epithelium in radial nerve cords, the arm tips, coelomocytes, and/or gonoduct acts on ovarian follicle cells within the ovaries. Continuously, 1-MeAde stimulates oocytes to produce maturation-promoting factor (MPF), which is a direct trigger for germinal vesicle breakdown (GVBD) and follicular envelope breakdown (FEBD). After the follicular envelope separates from the oocytes, the denuded mature oocytes can move freely within the ovary. 1-MeAde also induces acetylcholine (ACh) secretion. ACh activates the muscarinic ACh receptor in muscle cells of gonadal wall. The activated G protein stimulates the cation channel to produce an influx of Na^+^. Then, the action potential is altered by depolarization to activate the voltage-dependent calcium channel (VDCC). Simultaneously, Ca^2+^ influx occurs and begins contraction and peristalsis in muscles in the gonadal wall to discharge mature eggs from the ovaries.

**Figure 6 biomolecules-13-00781-f006:**
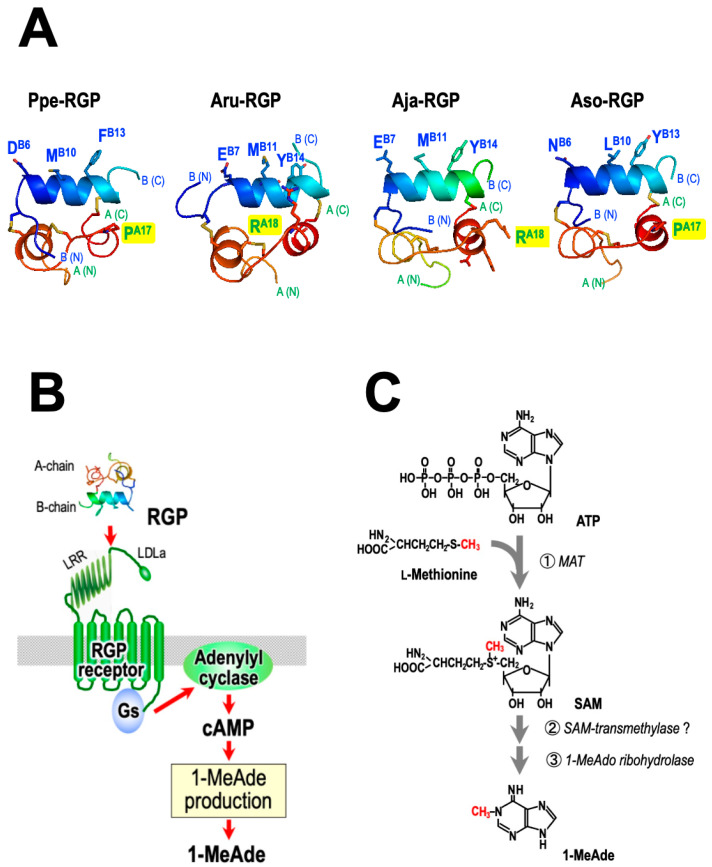
Action of RGP on ovarian follicle cells to induce 1-MeAde production. (**A**) Three-dimensional (3D) structure models of Ppe-RGP, Aru-RGP, Aja-RGP, and Aso-RGP. The side chains of selected amino acids in the A (green) and B-hain (blue) shown and labelled are possibly involved in binding to the receptor. Each 3D structure model was produced using SWISS-MODEL with the default setting (https://swissmodel.expasy.org, accessed on 16 January 2018 and 7 August 2019). The stereochemical quality values of the 3D models determined by ERRAT (https://saves.mbi.ucla.edu, accessed on 19 February 2023) were 81.5% for Ppe-RGP, 96.2% for Aru-RGP, 95.1% for Aja-RGP, and 86.4% for Aso-RGP. Thus, the 3D models are reliable. (**B**) Signal transduction of RGP in ovarian follicle cells. RGP acts on the RGP receptor (RGPR) in the surface of follicle cells. RGPR is a G protein-coupled receptor that harbors a low-density lipoprotein receptor class A (LDLa) motif and leucine-rich repeat (LRR) sequences in the extracellular domain of the N-terminal region. Upon the binding of RGP to RGPR, adenylate cyclase is activated via a Gs protein. As a result, 1-MeAde biosynthesis is activated by increasing intracellular cyclic AMP (cAMP) levels. (**C**) Possible metabolic pathway of 1-MeAde biosynthesis. The adenine moiety of 1-MeAde is derived from ATP, and the methyl group is from L-methionine. Presumably, S-adenosyl-methionine (SAM) has an important role in 1-MeAde biosynthesis. It is unclear whether the reaction system from SAM to 1-methyladenosine (1-MeAdo) is the precursor of 1-MeAde. ①: L-methionine adenosyltransferase (MAT) (EC 2.5.1.6); ②: SAM-transmethylase; ③: 1-MeAdo ribohydrolase (EC 3.2.2.13).

**Figure 7 biomolecules-13-00781-f007:**
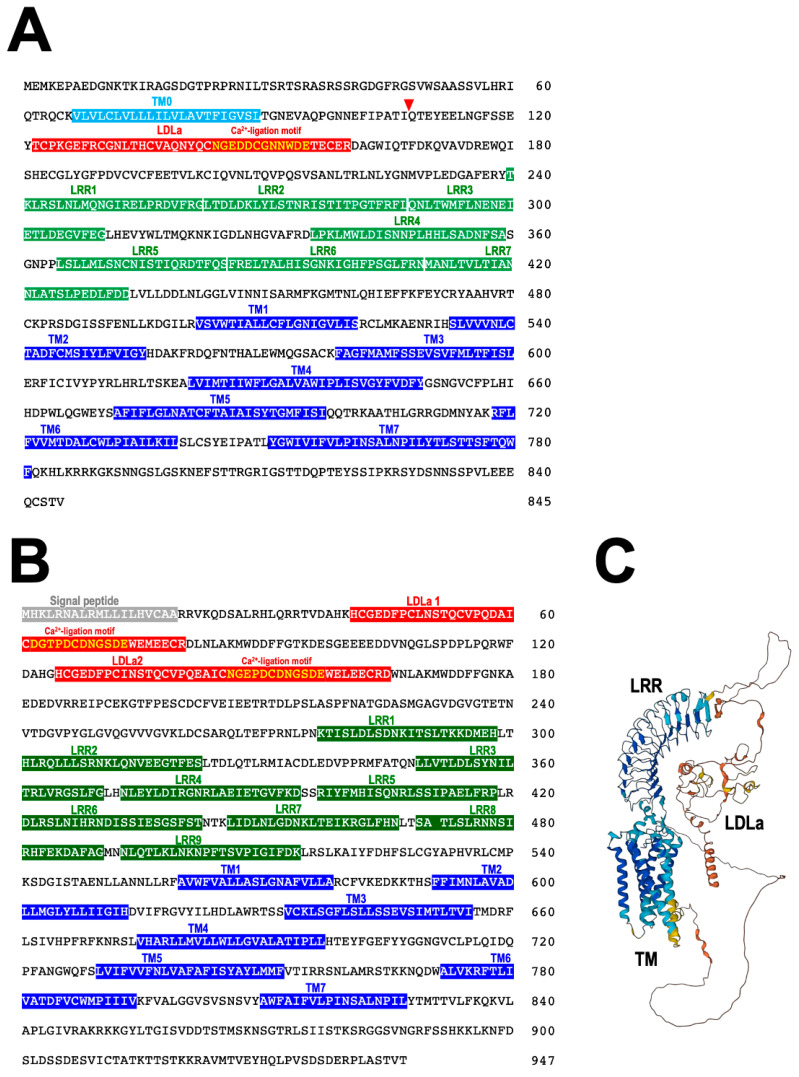
Protein sequences of putative RGP receptors (RGPR) and 3D structure model. Deduced protein sequences of *P. pectinifera* RGPR1 (Ppe-RGPR1) (**A**) and RGPR2 (Ppe-RGPR2) (**B**). The signal peptide, low-density lipoprotein receptor class A (LDLa), leucine-rich repeat (LRR), and transmembrane domains (TM) are indicated with gray, red, green, and blue backgrounds, respectively. Yellow characters indicate the Ca^2+^ ligation motifs. The inverted red triangle mark indicates the predicted internal cleavage site. (**C**) The 3D structure model of Ppe-RGPR2 is cited from the AlphaFold Protein Structure Database: UniProt A0A7G1HMC2 (https://alphafold.ebi.ac.uk/, accessed on 19 February 2023). The stereochemical quality value of the 3D model was 81.5%, so the model is reliable.

**Figure 8 biomolecules-13-00781-f008:**
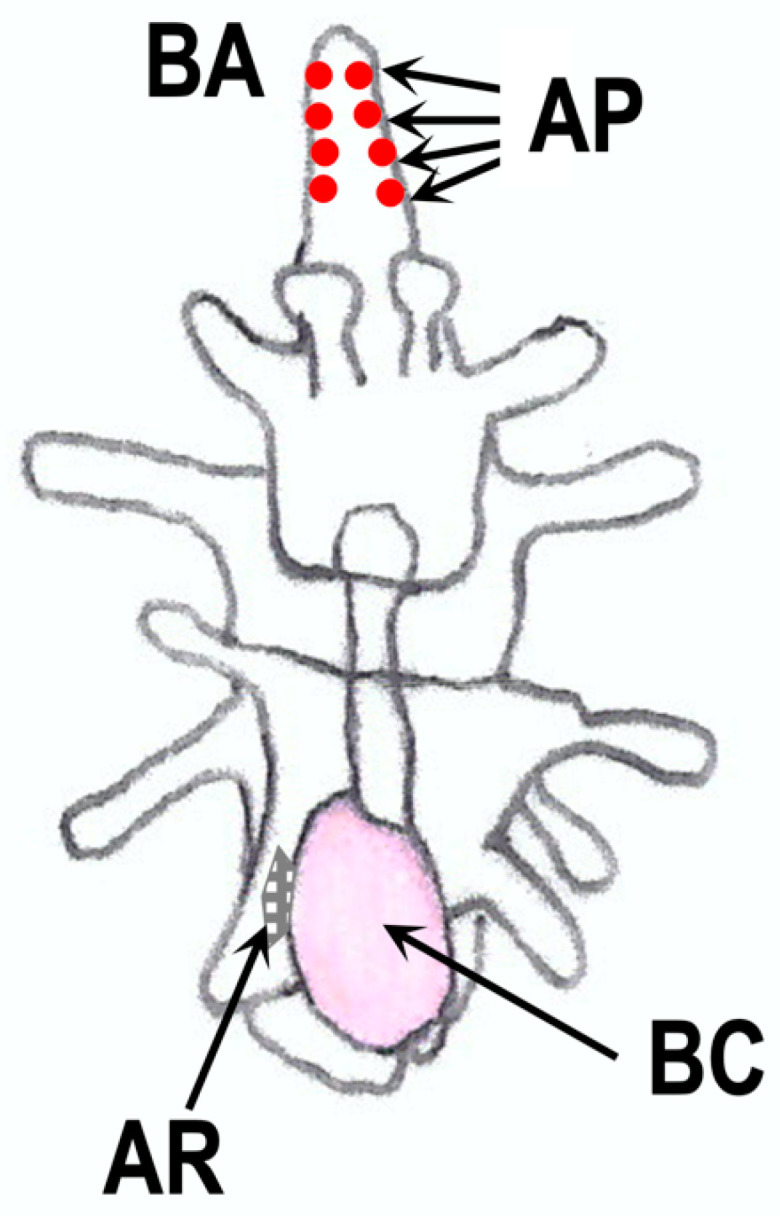
Schematic diagram of the localization of RGP in starfish brachiolaria larvae. AP, adhesive papilla; AR, adult rudiment; BA, brachiolaria arm; BC, body cavity.

## Data Availability

All datasets generated for this article are included in the manuscript and/or the Appendix A.

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
