# Peer review of "Relaxin-like Gonad-Stimulating Peptides in Asteroidea"

_biomolecules, 2023, doi:10.3390/biom13050781_

Round 1

Reviewer 1 Report

This review manuscript explained well the Relaxin-like gonad-stimulating peptides in Asteroidea class, most diverse and familiar class in the phylum echinodermata. The manuscript is well organized and written.  The manuscript need minor amendments before publication.

Minor comments:

Line 200-206: It would be appreciable if you use the accession number of incorporated sequence in figure 2.

Line 350: It is necessary to evaluate the stereo-chemical quality of the predicted mode using the protein quality predictor, verify 3D, and ERRAT.

Line 419: Please check the quality of 3D model.

Author Response

Line 200-206: It would be appreciable if you use the accession number of incorporated sequence in figure 2.

Response: Following the suggestion, RGPs with accession numbers are listed in Table S1. I have added the sentence to the legend of Figure 3 (Lines 222-223).

Line 350: It is necessary to evaluate the stereo-chemical quality of the predicted mode using the protein quality predictor, verify 3D, and ERRAT.

Response: The point is well taken. The stereo-chemical quality values of the 3D models of RGPs were determined by ERRAT. These values were around 90%, so the models seem reliable (Lines 372-374).

Line 419: Please check the quality of 3D model.
Response: The stereo-chemical quality value was 81.5% (Lines 447-448).

Reviewer 2 Report

Review comments for biomolecules-2264315

General comments

This review summarizes the work that has revealed the starfish's gonadotropin-like active substance, which has long been proposed as GSS but whose details have remained unclear. The author summarizes the structural features of the RGP molecule widely identified in starfish and proposes the existence of two types, RGP1 and RGP2, with the addition of a newly discovered, similar RGP2 peptide.

The distribution and function of RGP neurons are discussed in detail in order to clarify the function of RGP peptides. The processes leading to gamete ovulation by 1-MeAde which production is induced by the RGPs brought about by receptor-mediated signaling and the release of gametes in concert with contraction of the gonad wall by muscarinic ACh are described in detail.

In addition, the author mentions the possibility of non-reproductive functions of RGPs based on the early appearance of RGPs during larval development.

This review is a valuable paper that organizes in detail the results of reproductive endocrine research in invertebrate starfish. But, some revisions are indicated below for publication and should be adequately addressed.

Minor comments

1.    P1-L11; Invertebrate gene names should appear in lowercase italics, like “rgp gene”.

2.    P4-L138 & Figure 2A; twenty-four preprohormones of RGPs identified from 24 species are mentioned in the text, but twenty-three genes were shown in figure 2A. Is one sequence missed?

3.    P5-L164 & L166; “Lse RGP” cannot be found in Figure2A.

4.    P5- L166; “Lcse RGP” is correctly “Lse RGP”?

Author Response

  1. P1-L11; Invertebrate gene names should appear in lowercase italics, like “rgp gene”.

Response: Modified as suggested (Line 11).

  1. P4-L138 & Figure 2A; twenty-four preprohormones of RGPs identified from 24 species are mentioned in the text, but twenty-three genes were shown in figure 2A. Is one sequence missed?

        Response: In the revised manuscript, Figure 2 and Figure 3 have been replaced. The precursor of Lse-RGP had not been listed, because of the incomplete peptide sequence. In the revised manuscript, the partial sequence of Lse-RGP precursor has been listed in Figure 3A.

  1. P5-L164 & L166; “Lse RGP” cannot be found in Figure2A.

        Response: As mentioned above, Lse-RGP precursor has been added to Figure 3A.

  1. P5- L166; “Lcse RGP” is correctly “Lse RGP”?

        Response: Modified as suggested (Line 170).

Reviewer 3 Report

The author provides a comprehensive review of relaxin-like gonad-stimulating peptides (RGPs) in Asteroidea. In general, the overall writing is good. However, minor revisions are required before acceptance for publication.

Rewrite the Abstract with more details of the outline and purpose of this review.

Line 94: “pair” should be “pairs”.

Lines 96-97: Change “base” and “b” to “nucleotides” and “nt”, respectively.

Lines 98-105: The sentences can be marked with (Figure 2) for clarity.

Move Figures 2 and 3 for convenient reading.

Lines 214-215: There are 24 RGP orthologs (lines 141-142), but only 21 RGP precursors were used for construction of the phylogenetic tree. Why?

Figure 4: It is strange that no outgroup was used.

Lines 477-479: Rewrite this sentence.

Author Response

Line 94: “pair” should be “pairs”.

Response: Modified as suggested (Line 97).

Lines 96-97: Change “base” and “b” to “nucleotides” and “nt”, respectively.

Response: Modified as suggested (Line 98-99).

Lines 98-105: The sentences can be marked with (Figure 2) for clarity.

Response: In the revised manuscript, Figure 2 and Figure 3 have been replaced. I have rewritten the sentence as suggested (Line 104)

Move Figures 2 and 3 for convenient reading.

Response: As mentioned above, Figure 2 and Figure 3 have been replaced.

Lines 214-215: There are 24 RGP orthologs (lines 141-142), but only 21 RGP precursors were used for construction of the phylogenetic tree. Why?

Response: Because the precursor of Lse-RGP is a partial sequence, the phylogenetic tree was constructed with 23 RGP orthologs, except for Lse-RGP (Lines 232-233).

Figure 4: It is strange that no outgroup was used.

Response: In the revised manuscript, the unrooted phylogenetic trees have been reconstructed without outgroup (Figure 4).

Lines 477-479: Rewrite this sentence.

Response: I have rewritten the sentence (Lines 502-503).

Reviewer 4 Report

This review article collectively addresses relaxin-like gonad-stimulating peptides (RGP) that have been identified or predicted in over 20 different species of Asteroidea. The comparison of the amino acid sequences for the homologous prohormone led to construction of a phylogenetic tree in which sequences belonging to the same animal order are generally grouped together. Based on the assumption that the receptor for RGP has to be related to receptors for relaxin in human, the author surveyed the starfish genome and transcriptome database to propose a couple of candidate receptors for RGP. Considering the spatiotemporal pattern of RGP during embryonic development, the author suggested that RGP might also have a role independent of gonadotropin-like activity.

This is a highly focused review, and the author is the expert on the topic. As RGP is analogous to gonadotropin, the review may be of interest to those who study endocrinology and reproductive biology. Being a review, it does not call for much criticism except for some minor comments that the author may want to elaborate on.

1.      The manuscript introduces a novel second class of RGP, namely RDP2, but its direct relationship or comparison with RGP was rather vaguely described. How they are different or similar in functional roles and protein structure. Comparisons were made within each group, but not much between the groups. 

2.      In search of receptors for RGP, the only criterion used was the homology to human relaxin family receptors while there are probably over a hundred G protein-coupled receptors in the starfish genome. The author retrieved two candidates out of this in silico search, but the justification of this assumption and the approach was not much explained.

3.      The sentence in lines 65-66 apparently requires rephrasing because its meaning is misleading. Original text: “Generally, oocyte maturation means the process that immature oocytes stimulated by a hormone resume meiotic arrest and become fertilizable.” Suggested text (example):  “Generally, oocyte maturation means the process in which immature oocytes stimulated by a hormone overcome meiotic arrest and become fertilizable.”              

4.      Line 374, perhaps “harbored” should be changed to “harboring”.

5.      In the Reference, at least 36 out of 132 citations (over 27%) are the publications of the author himself. To extend to the comparable or related work of others and find a common ground, the reference should be diversified to have a more balanced bibliography.    

Author Response

  1. The manuscript introduces a novel second class of RGP, namely RDP2, but its direct relationship or comparison with RGP was rather vaguely described. How they are different or similar in functional roles and protein structure. Comparisons were made within each group, but not much between the groups. 

Response: The point is well taken. It is unknown why RLP2, which has a different protein structure from RGP, showed spawning-inducing activity. It will be necessary to clarify whether RLP2 acts on the same receptor as RGP and its physiological significance. (Lines 204-205). I can say is that RGP and RLP2 are paralogs of each other (Lines 252-253).

  1. In search of receptors for RGP, the only criterion used was the homology to human relaxin family receptors while there are probably over a hundred G protein-coupled receptors in the starfish genome. The author retrieved two candidates out of this in silico search, but the justification of this assumption and the approach was not much explained.

Response: I have rewritten the text on how to identify RGP receptors (Lines 390-394).

  1. The sentence in lines 65-66 apparently requires rephrasing because its meaning is misleading. Original text: “Generally, oocyte maturation means the process that immature oocytes stimulated by a hormone resume meiotic arrest and become fertilizable.” Suggested text (example):  “Generally, oocyte maturation means the process in which immature oocytes stimulated by a hormone overcome meiotic arrest and become fertilizable.”  

Response: I have modified the sentence as suggested (Lines 67-68).

  1. Line 374, perhaps “harbored” should be changed to “harboring”.

Response: Modified as suggested (Line 397).

  1. In the Reference, at least 36 out of 132 citations (over 27%) are the publications of the author himself. To extend to the comparable or related work of others and find a common ground, the reference should be diversified to have a more balanced bibliography.    

Response: Regarding gonadotropin-like active substances in invertebrates, I have also discussed about ascidians in addition to starfish (Lines 190-197).